# Impact of Freeze- and Spray-Drying Microencapsulation Techniques on β-Glucan Powder Biological Activity: A Comparative Study

**DOI:** 10.3390/foods11152267

**Published:** 2022-07-29

**Authors:** Veronika Valková, Hana Ďúranová, Aude Falcimaigne-Cordin, Claire Rossi, Frédéric Nadaud, Alla Nesterenko, Marvin Moncada, Mykola Orel, Eva Ivanišová, Zuzana Chlebová, Lucia Gabríny, Miroslava Kačániová

**Affiliations:** 1AgroBioTech Research Centre, Slovak University of Agriculture, Tr. A. Hlinku 2, 94976 Nitra, Slovakia; veronika.valkova@uniag.sk (V.V.); hana.duranova@uniag.sk (H.Ď.); mykola.orel@uniag.sk (M.O.); zuzana.chlebova@uniag.sk (Z.C.); lucia.gabriny@uniag.sk (L.G.); 2Enzyme and Cell Engineering, UPJV, CNRS, Université de Technologie de Compiègne, Centre de Recherche Royallieu-CS 60319-60 203 CEDEX, 60200 Compiègne, France; aude.cordin@utc.fr (A.F.-C.); claire.rossi@utc.fr (C.R.); 3Service d’Analyse Physico-Chimique, Université de Technologie de Compiègne, Centre de recherche Royallieu-CS 60319-60 203 CEDEX, 60200 Compiègne, France; frederic.nadaud@utc.fr; 4Integrated Transformations of Renewable Matter, ESCOM, Université de Technologie de Compiègne, Centre de Recherche Royallieu-CS 60319-60 203 CEDEX, 60200 Compiègne, France; alla.nesterenko@utc.fr; 5Department of Food, Bioprocessing, and Nutrition Science, Nord Carolina State University, Raleigh, NC 27606, USA; mlmoncad@ncsu.edu; 6Institute of Food Sciences, Slovak University of Agriculture, Trieda Andreja Hlinku 2, 94976 Nitra, Slovakia; eva.ivanisova@uniag.sk; 7Institute of Horticulture, Faculty of Horticulture and Landscape Engineering, Slovak University of Agriculture, Tr. A. Hlinku 2, 94976 Nitra, Slovakia; 8Department of Bioenergy, Food Technology and Microbiology, Institute of Food Technology and Nutrition, University of Rzeszow, 4 Zelwerowicza Str., 35-601 Rzeszow, Poland

**Keywords:** microencapsulation, freeze-drying, spray-drying, maltodextrin, β-glucan, SEM, DPPH assay, biologically active substances

## Abstract

The study compares the impact of freeze- and spray-drying (FD, SD) microencapsulation methods on the content of β-glucan, total polyphenols (TP), total flavonoids (TF), phenolic acids (PA), and antioxidant activity (AA) in commercially β-glucan powder (*Pleurotus ostreatus*) using maltodextrin as a carrier. Morphology (scanning electron microscopy- SEM), yield, moisture content (MC), and water activity (a_w_) were also evaluated in the samples. Our examinations revealed significant structural differences between powders microencapsulated by the drying methods. As compared to non-encapsulated powder, the SD powder with yield of 44.38 ± 0.55% exhibited more reduced (*p* < 0.05) values for a_w_ (0.456 ± 0.001) and MC (8.90 ± 0.44%) than the FD one (yield: 27.97 ± 0.33%; a_w_: 0.506 ± 0.002; MC: 11.30 ± 0.28%). In addition, the highest values for β-glucan content (72.39 ± 0.38%), TPC (3.40 ± 0.17 mg GAE/g), and TFC (3.07 ± 0.29 mg QE/g) have been detected in the SD powder. Our results allow for the conclusion that the SD microencapsulation method using maltodextrin seems to be more powerful in terms of the β-glucan powder yield and its contents of β-glucan, TP, and TF as compared to the FD technique.

## 1. Introduction

To improve human health in association with energy intake, modulation of metabolic and inflammatory processes, oxidative stress (antioxidant activity; AA), enzymes and receptor activities, and many others, the intake of bioactive compounds in foods is of primary importance [1,2]. Of them, β-glucan is a water-soluble dietary fiber commonly found in oats, barley, bacteria, yeast, algae, and mushrooms [3]. Structurally, β-glucans are a heterogeneous group of glucose polymers with a common structure comprising a main chain of β-(1,3) and/or β-(1,4)-glucopyranosyl units known especially for the immunomodulatory and anti-inflammatory properties [4]. Additionally, β-glucans may protect the cardiovascular system via ameliorating glucose, lipid metabolism, and blood pressure, and another promising application concerns cancer disorders as an adjuvant of conventional chemotherapy [5]. According to Herrera et al. [6], the β-glucan content and its physicochemical properties (such as solubility and viscosity) are principal parameters, not only in providing human health benefits, but also in evaluating its impact on the processing and sensory properties of foods when used to produce food products with added nutritional value. Owing to the occurrence of biologically active substances, oyster mushrooms (*Pleurotus* species) possess a plethora of bioactive properties including hypocholesterolemic, antioxidant, anti-bacterial, anti-diabetic, hepatoprotective, anti-carcinogenic, anti-viral, anti-arthritic, and immune-modulatory ones [7]. In addition to β-glucan [8], *Pleurotus* (*P*.) *ostreatus* as a main edible mushroom also contains different types of phenolic compounds, including phenolic acids and flavonoids [9], and exhibits good AA [10].

To preserve the biological activity and improve the stability of the bioactive compounds as well as ensure the controlled release of the latter [11], their encapsulation is among the most popular alternatives in food processing [12]. Indeed, this process of applying relatively thin coatings of solids, liquid droplets, or even gas bubbles (termed as wall or shell material, coating, or carrier agent) around a substance (active or core material) [13,14] has the advantage of being a nonthermal stabilization approach suitable for temperature-sensitive natural biologically active substances such as those extracted from different matrices with medical applications [11]. In addition to the food industry [15,16,17], encapsulation is also employed in biomaterials, tissue bio-nanoengineering [18] and biology [19], pharmacy [20,21], agriculture [22,23], and the cosmetic industry [24]. Based on the size of the encapsulated particles, nanoencapsulation (10–1000 nm) and microencapsulation (3–800 µm) can be distinguished [25]. Regarding microencapsulation, there is a wide range of different microencapsulation techniques applied for bioactive compounds such as freeze-drying [26,27,28,29,30], spray-drying [26,27,28,29,30,31,32], ionic gelation [31,33,34], double emulsion solvent evaporation [35], molecular inclusion in cyclodextrin [36,37], ionic crosslinking method [38,39,40], electrospinning [41,42,43], and spray-chilling [44], all of them having some advantages and disadvantages. Among them, SD is one of the most widely used microencapsulation techniques in the manufacturing of food ingredients at a large scale due to a wide range of coating material options, rapid water evaporation, and the possibility to control the temperature to avoid any degradation of the product. Prior to SD, a liquid formulation (suspension, emulsion, or solution) containing the bioactive compound and coating material is prepared [45]. Then this mixture is transformed into a powder by its spraying into a hot drying gas and fast solvent evaporation. This method is commonly applied for the protection of active ingredients and heat-sensitive compounds [46], resulting in powders with high quality and stability, low water activity (a_w_), and easier transport and storage [47]. It is extensively used in the food industry due to its simplicity, low cost, reproducibility and ease to scale up [48], and it is effectively and highly preferred due to economic advantages compared to other techniques of microencapsulation [49]. In contrast, its main limitation with respect to the other preserving processes (such as FD) is the loss of volatiles (since they can diffuse within the liquid phase during drying, or even into the solid crust after drying). Moreover, care must be taken with a regard to the inlet and outlet temperatures, since excessively high temperatures can cause degradation of some product components [47]. Another ordinary problem of this technique is the sticking of the processed material to the walls of the drier, which can reflect the deterioration of the product yield [50]. On the other hand, the FD technique is simpler than the other ones because of the limited number of steps [30], first inducing the freezing of water, followed by its removal from the sample, initially by sublimation (primary drying at low shelf temperature and a moderate vacuum), and then by desorption (secondary drying facilitated by raising shelf temperature and reducing chamber pressure to a minimum) [51]. Although it requires a much longer time, it can be performed at a very low temperature [52] which is another advantage of this method. Importantly, the selection of suitable carrier agents for microencapsulation is another principal step to ensure the technique’s efficiency. As a wall material, maltodextrin (MD) is preferentially applied in bioactive compound encapsulation because of its low cost, favorable biocompatibility, moldability, and physicochemical properties (mainly high solubility and low viscosity), effectively preventing microparticles aggregation [47,53].

The current study was designed to evaluate and compare microparticle structures and alterations in powder yield, MC, a_w_, the presence of selected biologically active compounds (TPC, PAC, FC), and AA produced by two different microencapsulation techniques (SD and FD) to choose the best one for further research into food products with this β-glucan powder addition. Taking into account the fact that β-glucan is usually employed in experiments as a coating material, our study as a first report can illuminate this unexplored research area, and the new knowledge can bring innovation to the scientific community on a large scale.

## 2. Materials and Methods

### 2.1. Materials

Powder extract of β-glucan (*P. ostreatus*) was produced by a commercial producer Dimenzia Ltd. (Kežmarok, Slovakia). The extract was microencapsulated using two different methods (FD, SD), thereby taking into consideration three samples to be evaluated: control sample (non-encapsulated β-glucan powder), Treatment 1 (powder encapsulated by FD), and Treatment 2 (powder encapsulated by SD).

Maltodextrin (MD; DE 13–17, Sigma-Aldrich, Saint-Quentin-Fallavier, France) was used as a coating material in both microencapsulation techniques applied. To prepare the MD solutions, ultrapure water (Milli-Q^®^ system, 18.2 MΩ/cm, Merck Millipore, Molsheim, France) was used.

Other chemicals employed were of analytical grade, and they were purchased from Sigma Aldrich (Sigma-Aldrich Chemie GmbH, Steinheim, Germany) or Megazyme (Megazyme, Bray, Co. Wicklow, Ireland).

### 2.2. Preparation of Biopolymer Solutions for β-Glucan Concentrate Encapsulation

First, the MD solution was prepared by dissolving it in ultrapure water in 20% (*w*/*v*) concentration. Then, an active β-glucan concentrate was added in 0.5% (*w*/*w* of MD) concentration, and the final liquid formulation was magnetically stirred for 30 min.

### 2.3. Freeze-Drying Method

Microencapsulation by FD was performed according to the method by Šeregelj et al. [54]. Briefly, before the lyophilization, the MD solution containing β-glucan concentrate was frozen in a freezer at −80 °C for 1 h. The frozen formulation was dried in a freeze-dryer (Heto Drywinner, Avantec, Paris, France) equipped with a vacuum pump and a condenser for 48 h. The obtained aerogel was crushed and homogenized into powders and finally stored in a transparent plastic bottle at room temperature (20 °C) until further analyses.

### 2.4. Spray-Drying Method

Microencapsulation by SD was realized according to the protocol described in previous studies [55,56]. A freshly prepared liquid formulation of β-glucan concentrate and MD was spray-dried using a lab-scale Mini Spray Dryer B-290 (Büchi, Flawil, Suisse) equipped with a nebulizer having an internal diameter of 0.7 mm and using compressed air at a controlled flow rate. During the SD process, the solution was stirred continuously in a magnetic stirrer at room temperature to ensure the solution homogeneity and was introduced into the dryer via a peristaltic pump at a constant flow of 360 mL/h. The drying air flow was fixed to 536 L/h and sprayed co-currently to the feeding solution. The SD was operated at 120 °C and 65 °C inlet/outlet temperatures, respectively. Finally, the SD microparticles were separated from the humid air in the cyclone, and the powder of microparticles was recovered in the collector. After the drying process, the powder was stored in a transparent bottle at room temperature (20 °C) until further analyses.

### 2.5. Determination of Product Yield

The product yield (Y) of the FD and SD powders was established for each system studied and was expressed as a ratio of obtained powder (M1) reported to the introduced amount of solid raw materials (M2) applied according to Equation (1):Y (%) = (M1/M2) × 100(1)

### 2.6. Evaluation of Moisture Content and Water Activity

Water activity (a_w_) was measured using a Lab Master a_w_ Standard (Novasina, Lachen, Switzerland). For this purpose, 1.0 g of each sample was placed into a sample pan. The value of a_w_ was measured automatically at 25 °C for 10–15 min.

Moisture content (MC) was determined using the moisture analyser DBS 60-3 (Kern & Sohn, Balingen, Germany). In this case, 1.0 g of each sample was weighed on the sample plate, and measurement was performed at 120 °C for 5–10 min.

### 2.7. Scanning Electron Microscopy

The surface morphology and size of the FD and SD microparticles were investigated by scanning electron microscopy (SEM) using a Quanta 250 FEG microscope (Thermofischer, Eindhoven, The Netherland). A thin layer of the powders was deposited on carbon tape, and the samples were consequently recovered by a thin palladium layer (0.5 nm) prior to analysis. The SEM images were acquired with a 20 kV accelerating voltage using the hivac mode.

### 2.8. Determination of Antioxidant Activity and the Contents of Biologically Active Compounds

Prior to analyses, ethanolic extracts were prepared from all investigated samples. For each extraction, 0.2 g of each sample was extracted by 20 mL of 80% ethanol for 2 h and centrifuged at 4000× *g* for 10 min in Rotofix 32A (Hettich, Spenge, Germany). To determine AA, total polyphenols content (TPC), phenolics acid content (PCA), and flavonoids content (FC) obtained supernatants were used.

#### 2.8.1. Evaluation of Antioxidant Activity

Antioxidant capacity of the samples was measured with 2,2-diphenyl-1-picrylhydrazyl (DPPH) radical; the procedure has been previously described by Valková et al. [57]. In brief, a volume of 0.4 mL of each sample extract was added to 3.6 mL of DPPH solution (0.025 g DPPH in 100 mL ethanol). Absorbance of the reaction mixture at 515 nm was determined using a Jenway 6405 UV/Vis spectrophotometer (ColeParmer, Stone, United Kingdom). The scavenging activity was calculated as a percentage (AA%) and determined according to Equation (2), where A0 is the absorbance of the control reaction (DPPH radical); A1 is the absorbance of the tested sample. The AA values increased in the following manner: weak (0–29%) < medium-strong (30–59%) < strong (60% and more).
AA% = [(A0 − AAT)/A0 × 100](2)

#### 2.8.2. Total Polyphenols Content

Total PC was measured by the method of Valková et al. [57] using the Folin Ciocalteu reagent. In this case, 0.1 mL of each extract was mixed with 0.1 mL of Folin Ciocalteu reagent, 1 mL of 200 g L^−1^ sodium carbonate, and 8.8 mL of distilled water. After 30 min in darkness, the absorbance at 700 nm was measured using a Jenway 6405 UV/Vis spectrophotometer (ColeParmer, Stone, UK). Gallic acid was used as a standard, and the results were expressed in milligrams of gallic acid equivalents (GAE) per g of dry weight (dw). The calibration curve followed Equation (3).
y = 0.0009x + 0.0012R^2^ = 0.9978(3)

#### 2.8.3. Phenolics Acids Content

The content of PA was determined using a previously published method [57]. A volume of 0.5 mL of each sample extract was mixed with 0.5 mL of 0.5 mol L^−1^ hydrochloric acid, 0.5 mL Arnow reagent (100 g L^−1^ sodium nitrite and 100 g L^−1^ sodium molybdate), 0.5 mL of 1 mol L^−1^ sodium hydroxide, and 0.5 mL of distilled water. Absorbance at 490 nm was measured using a Jenway 6405 UV/Vis spectrophotometer (ColeParmer, Stone, UK). Caffeic acid was used as a standard, and the results were expressed in milligrams of caffeic acid equivalents (CAE) per gram of dw. The calibration curve followed Equation (4).
y = 0.0051x + 0.0064R^2^ = 0.9996(4)

#### 2.8.4. Flavonoids Content

Flavonoids content was determined by the method of Ivanišová et al. [58]. A quantity of 0.5 mL of each sample was mixed with 0.1 mL of 10% (*w*/*v*) ethanolic solution of aluminum chloride, 0.1 mL of 1 M potassium acetate, and 4.3 mL of distilled water. After 30 min in darkness, the absorbance at 415 nm was measured using a Jenway 6405 UV/Vis spectrophotometer (ColeParmer, Stone, UK). Quercetin was used as a standard. The results were expressed in milligrams of quercetin equivalents (QE) per gram of dw. The calibration curve followed Equation (5).
y = 0.0021x + 0.0229R^2^ = 0.9977(5)

#### 2.8.5. Determination of β-Glucan Content

The β-glucan content in the samples was determined using the enzymatic kit β-glucan Assay kit (Mushroom and Yeast) K-YBGL (Megazyme, Bray, Co. Wicklow, Ireland). The β-glucan determination procedure was implemented in accordance with the manufacturer’s methodical instructions that are included in the kit, and all instructions are listed in Mushroom and Yeast Beta-Glucan Assay procedure K-YBGL 11/19, with our minor methodical optimization. Instead of 12 M, H_2_SO_4_ (H_2_SO_4_ concentration 98%, sp. gr. 1.835), 18 M H_2_SO_4_ (H_2_SO_4_ concentration 96%, sp. gr. 1.835) was used. The methodological procedure’s accuracy was also determined within our analyses by using the control yeast β-glucan preparation (49% of β-glucan content) as an enclosed substance of the analytical kit. Various appliances were used during the research work, including laboratory scales KERN ABT 220-DM (KERN & SOHN GmbH, Balingen, Germany); vortex mixer bioSan V-1 plus, Personal Vortex (bioSan, Riga, Latvia); water bath Memmert U 1.28 (Memmert GmbH + Co. KG, Schwabach, Germany); centrifuge Hettich Mikro 185 (Andreas Hettich GmbH & Co. KG, Tuttlingen, Germany); and spectrophotometer UV/Vis Cary 60 (Agilent, Santa Clara, CA, USA).

### 2.9. Statistical Analysis

For statistical comparison of our data, software GraphPad Prism 8.0.1 (GraphPad Software Incorporated, San Diego, CA, USA) and One-way ANOVA were applied. In statistically significant results (*p* < 0.05), the Tukey HSD post hoc test was performed. All determinations were performed in triplicate.

## 3. Results

### 3.1. Sample Size and Morphology

Morphological analysis of both FD and SD samples was conducted by scanning electron microscopy (SEM). Using MD as a coating material, both microencapsulation techniques of the originally fine-grained, slightly yellow powder resulted in white-colored powders with finer structures being the finest one in the SD samples. Scanning EM displayed that FD (Figure 1) led to the formation of irregular-shaped glassy particles with a size larger than 100 µm showing some pores on their surfaces (red arrow in Figure 1A). The structures of the powder seemed to be rather brittle and uneven, without any evidence of the structural collapse related to stickiness at the macroscopic level. In addition, budlike spherical domains (yellow arrow in Figure 1C) and scaly structures (blue arrow in Figure 1B) were present on their surfaces. By contrast, SD microparticles of β-glucan powder (Figure 1D–F) showed more or less spherical conformation in various sizes of individual microparticles ranging from 2–3 to 20–30 µm, approximately. This result is typical for microparticles obtained by spray-drying. The dented and wrinkled surface of the particles is due to the rapid formation of an interfacial crust followed by the evaporation of a high amount of water during the SD process. The smooth shape without apparent pores, cracks, fissures, or interruptions can be observed, suggesting efficient protection of the active core inside the microparticles.

### 3.2. Yield of Microencapsulated Samples

Powder yield was significantly (*p* < 0.05) influenced by the drying method employed. Indeed, 44.38 ± 0.55% of the yield was produced by the SD technique, while the FD one led to a lower value (27.97 ± 0.33%) for the powder yield.

### 3.3. Moisture Content and Water Activity of Samples

The data from a_w_ and MC analyses is shown in Table 1. From that, a significant impact of both microencapsulation techniques on the physical parameters of the β-glucan extract powder can be seen. In this line, the SD method seems to be more efficient in decreasing a_w_ and MC of the powder, indicating its higher microbial stability as compared to that obtained using the method of FD.

### 3.4. β-Glucan Content of Samples

Table 2 summarizes the content of β-glucan in all experimental samples. The results of the study indicate that the β-glucan content was higher in the experimental samples which were prepared using the FD and SD methods as compared to the control sample. A beneficial effect of both microencapsulation techniques on the measured parameter of the β-glucan powder was more pronounced in the SD one.

### 3.5. Antioxidant Activity and Total Polyphenols, Phenolics Acids, and Flavonoids Contents of Samples

Both powders produced by the FD and SD methods using MD as an encapsulant showed AA preservation of the β-glucan powder. In addition, TPC, PAC, and FC were retained in these powders. Moreover, the contents of TP and flavonoids in the SD samples were significantly higher (*p* < 0.05) than those determined in the non-encapsulated (control) ones (Table 3).

## 4. Discussion

Owing to the ability of bioactive compounds to stabilize via the structuring of systems that preserve their chemical, physical, and biological properties, and their release (or delivery) under desired conditions [59], microencapsulation technologies have been gaining popularity in the food industry during the last few decades. Although β-glucan is usually used as a wall agent (encapsulant) for bioactive compound encapsulation [60,61,62,63,64,65], this soluble fiber was employed in our study as a core material to be microencapsulated using two different techniques to improve its stability and nutritional value for scientific purposes and practical applications.

Since the protection capacity of polymeric wall materials in the encapsulated products is associated with the degree of porosity and integrity of individual microparticles [66,67], the investigation of their microstructure is a crucial step in assessing the effectiveness of encapsulation technique(s) and carrier agent(s). Our results showed that both drying methods (freeze and spray) resulted in the formation of microparticles with completely distinctive morphological characteristics which can affect their solubility in water [27]. Supporting the findings of the current study, the spherical shape of the different-sized microparticles with concavities in SD powders, along with the irregular, brittle, and porous structures of the FD powders obtained from acerola (*Malpighia emarginata* DC) pulp and saffron (*Crocus Sativus* L.) petal phenolic extracts have also been visualized by SEM in the scientific research of Rezende et al. [66], and Ahmadian et al. [28], respectively. Generally, the size and morphology of SD microparticles depend on the different formulation and process-related parameters, such as inlet and outlet temperatures, feeding rate, initial concentration of the solid material, surface tension, and the intrinsic properties of the drying matrices [46]. In this context, lower inlet temperatures and a higher extract feed rate are, for instance, responsible for creating a product with higher particle size and agglomerated appearance, which could be attributed to higher residual MC as a consequence of the atomization conditions [68]. The spherical conformation of our SD microparticles is linked to the spraying-induced production of droplets which are consequently converted into the spherical particles by solvent evaporation [66]. On the other hand, the microparticle surfaces with concavities or indentations observed in our SD samples are most likely related to the liquid drop shrinkage because of rapid and drastic evaporation at the early stages of the SD process when the droplets enter the drying chamber, followed by cooling of the microparticles [26,69]. Most importantly, the integrity of these surfaces was continuous (not interrupted by any cracks) in our SD samples, which is an essential attribute to ensure lower oxygen permeability, better protection, and retention of core materials [28,70]. In line with our findings, erythrocyte-like surfaced microparticles were also observed in the samples of *Moringa oleifera* leaf extract microencapsulated with MD by the SD method [71]. Moreover, agglomeration of microparticles in SD powders encapsulated with MD is correlated with the carrier concentration since MD is capable of high humidity absorption, increasing cohesion of the microparticles [28]. Based on this aspect and our finding considering the absence of agglomeration in our SD powder, we assume that the dose of MD used for our microencapsulation process was appropriately selected. In contrast, the amorphous material produced by our FD microencapsulation displayed a sharp and more structured porous morphology with budlike spherical domains compared to microparticles obtained by the SD method. Similar morphology was also observed by González-Ortega et al. [29] in FD-encapsulated olive leaf extract. The authors noted that the ice sublimation which gives rise to a structure created by a glassy matrix with air cells (whose shape and size are related to the processing conditions employed and initial system composition) is the main cause of the high porosity observation. This process commonly plays a key role in preventing shrinkage and collapse of the structure causing an insignificant alteration in its volume [66]. Ultimately, very low temperature and processing under vacuum conditions (putting the particles under pressure) [28], and the lack of the strength to break the frozen droplets or to modify the surface during the drying process [66] are responsible for the scaly structure and larger size of our FD microparticles. According to González-Ortega et al. [29], the initial cell-like structure and wall characteristics of FD systems are also partly preserved in powders following the grinding. To our best knowledge, our report seems to be the first one concerning the morphology and size of FD and SD β-glucan microparticles encapsulated using MD. For this reason, the values for the size of our SD microparticles could not be directly compared to other studies.

Powder production yield belongs to the most important indicator showing the efficiency of the drying process [72] and significantly affecting the cost [73]. Despite several studies revealing a higher yield of FD microencapsulated samples compared to the SD ones [26,50], an opposite phenomenon was noted in our study. In effect, the yield of our SD treated powder was almost 17% higher than that produced by FD. In line with our results, Tonon et al. [74] recorded that the yield of açai fruit powder microencapsulated by the SD technique using MD as carrier did not exceed 48.4%. Similarly, SD microencapsulation of noni fruit powder with different concentrations of MD resulted in a yield not higher than 48.1% [75]. Based on the findings, any extensive losses of our powder because of the microparticle sticking on the walls of the SD drier and cyclone itself can be excluded. Hence, the assumption of precise removal of powder from the SD dryer is considered. Generally, MC plays a principal control point in the determination of food quality and preservation; however, foodstuffs with the same level of MC can differ considerably in safety and spoilage susceptibility [76]. Thus, MC should be parallel measured with a_w_ to provide a complete moisture analysis [77]. Indeed, since a_w_ describes the energy state of water in food, it has an ability to act as a solvent participating in both chemical/biochemical (oxidation and enzymatic) reactions and microorganism growth [78]. So, reduction of a_w_ in foods commonly increases the lag phase of microorganisms and declines the growth rate of vegetative microbial cells, germination of spores, and toxin production by molds and bacteria [79], prolonging the product shelf-life [80]. According to Erkmen and Bozoglu [79], dried or low-moisture foods do not contain more than 25% moisture and a_w_ < 0.60 (ensuring microbial stability), which is in accordance with our findings considering non-encapsulated (control) β-glucan powder. In addition, its value for a_w_ (0.51 ± 0.001) was in the range of those obtained from *P*. *sajor-caju* dried using sun (0.57 ± 0.02), a lab oven (0.57 ± 0.01), and low heat air blow (0.49 ± 0.03) [81]. To preserve β-glucan in its physiologically active state, the low a_w_ of the product during various kinds of processing (e.g., encapsulation) is critical [82]. In accordance with our findings concerning both microencapsulated powders, similar values for a_w_ were reported in FD powders prepared from an anthocyanin-rich extract of a blackberry by-product [80], an SD Gac fruit aril [83], and raspberry juice powders [84], all of them using MD as a wall material for encapsulation. In addition, Schmitz-Schug et al. [85] have determined MC and corresponding a_w_ of SD infant formula produced in a laboratory scale to be between 3.7 (± 0.6) and 9.9 (± 1.1)% wet basis (wb) and between 0.10 (± 0.01) and 0.85 (± 0.04), respectively, being in line with our study. Likewise, Tangarife et al. [86] have found MCs of SD powdered products to vary between 1.50% and 9.55%. Low MC values are commonly necessary to ensure powder stability in the concept of particle agglomeration, powder flow, and dispersion [70,87]. In this context, MC of our control β-glucan powder was revealed to be positively influenced by both drying techniques employed. Moreover, the SD method exhibited more effectiveness (*p* < 0.05) in reduction of the powder a_w_ and MC than the FD one, suggesting its outstanding greater increase in product stability. Similarly, Daza et al. [87] have reported significantly lower values for a_w_ and MC in SD samples of *Eugenia dysenterica* extract powder compared to those dried by freezing. We propose that this higher effectiveness of the SD method (than FD) in the context of declining powder MC can be associated mainly with the parameters of SD processing, particularly with high temperature which has a marked impact on MC of powders. Indeed, an increase in inlet temperature during the SD process results in an increased rate of water evaporation through enhanced heat transfer between air and droplets [68,88], rapidly decreasing MC of encapsulated matrices. Interestingly, MC values of our FD samples were lower compared to the non-encapsulated (control) powder, whilst those for a_w_ were considerably (*p* < 0.05) higher. In effect, there is an overall trend that a_w_ almost always increases with increasing MC, but the relationship is not linear. However, at a few distinct points, MC becomes lower as a_w_ increases [89]. According to Oberoi and Sogi [90], this inverse relationship between MC and a_w_ might be due to the high percentage of amorphous sugars with high hygroscopicity absorbing more free water from ambient air. Increased a_w_ of our FD powder can be attributed to the higher water retention of MD due to the high number of ramifications with hydrophilic groups [47] or higher powder porosity facilitating penetration of water from ambient air into the pores [91] during powder handling after the drying. Anyway, having the a_w_ lower than 0.6 [80,92], the powders produced by both microencapsulation techniques used in the current study can be considered enzymatically and microbiologically stable.

It is well-known that fungal β-glucan has a stimulating action on the human immune system with proven effects in the treatment of different illnesses [93]. Hence, part of our experiment was focused on the determination of its content in all our samples to evaluate its retention performed by both microencapsulation techniques. Sari et al. [94] determined β-glucan content in nine culinary mushrooms (including *Lentinula edodes* Berk. and five *Pleurotus species*) to range from 15 to 22% per dry matter. Similarly, Khan et a. [95] have found the content of β-glucan in mushroom samples of *Agaricus bisporus*, *P*. *ostreatus*, and *Coprinopsis atramentarius* to be 8.51 ± 2.45%, 14.18 ± 0.57%, and 17.02 ± 0.95%, respectively, which is contrary to our control displaying its higher content (58.78 ± 0.54%). This fact can indicate a better β-glucan extraction technique from oyster mushrooms; however, we must also keep in mind other factors (mainly growing conditions) contributing to this discrepancy between the studies. Additionally, significantly (*p* < 0.05) higher values for β-glucan content in the powders have been achieved by their microencapsulation using both drying methods as compared to the control one. Furthermore, this effect on the content of β-glucan was again shown to be more obvious in the SD samples than in those dried by freezing which is in line with the study conducted by Karthik and Anandharamakrishnan [96] showing the SD technique to be more efficient in the preservation of biologically active compounds in comparison with other employed encapsulation methods. Since MD is a hydrophilic polysaccharide consisting of D-glucose units linked primarily by α-(1 → 4) glycosidic linkages [97,98], the increase in the β-glucan concentration in our microencapsulated samples can be related to their unknown interactions. In this sense, it could be hypothesized that glucose units of the MD might interact with β-glucan functional groups of our powder during coating processes, thus increasing the β-glucan content in the dried samples.

Between both powders produced by two different microencapsulation techniques used in our study, no significant differences in AA, TPC, PAC, and FC have been observed, suggesting their same effectiveness with respect to the bioactive compounds’ preservation. On the other hand, based on the findings of demonstrably higher (*p* < 0.05) contents of TP and TF in our SD samples than in the non-encapsulated (control) ones, a greater power of the SD technique to stabilize these bioactive compounds of the β-glucan powder compared to that of the FD one must be taken in consideration. In effect, microencapsulation by SD is widely employed to protect polyphenols in various matrices [70,99,100,101,102,103,104,105], and the ability was also confirmed in our study. Greater efficacy of SD on increasing content of TP and TF identified in our samples can again be associated with the MD which not only preserved the content of the phenolic compounds in both microencapsulated formulations but also most likely had a higher affinity to polyphenols and flavonoids during the extraction in SD microencapsulated samples. Indeed, it has been postulated that polyphenols are able to bind to polysaccharides; hydrogen bonds are formed between hydroxyl groups of polyphenols and oxygen atoms of the glycosidic linkages of polysaccharides, and covalent bonds can be formed between phenolic acids and polysaccharides [106]. In this regard, for instance, the study performed by Triyono et al. [107] showed higher FC in *Physalis angulata* L. leaf extract obtained using the addition of various concentrations of MD to aqueous or ethanol solvents in comparison with those solely extracted by aqueous or ethanol solvents. Maltodextrin as a convenient encapsulating agent for polyphenol compounds obtained from diverse natural plant matrices has been shown in many reports [47,75,101,105,108,109,110,111]. Furthermore, it was demonstrated that the binding of polyphenols to polysaccharides depends on the properties of the polyphenol structure, such as molecular weight, flexibility of chemical structure, and hydroxyl group numbers [111]. Based on the mentioned aspects, we can hypothesize that the SD microencapsulation could positively influence the structure of polyphenols to a great extent (to produce their higher affinity to MD resulting in their greater extraction) compared to the FD method, which was demonstrated by significant differences in TPC and FC identified only between non-encapsulated and SD samples (none in FD samples). The values for TPC and FC estimated in our non-encapsulated β-glucan powder were in the range of those (TPC: 2.55–4.05 mg/g) detected in dry commercial *P*. *eryngii* isolates [112] and similar to that (FC: 2.11 mg/g of extract) found in *P. ostreatus* extract [9]. Taking into consideration AA, the DPPH scavenging ability of our non-encapsulated β-glucan powder was similar to those extracted from *P. ostreatus* [95], yeast (*Saccharomyces cerevisiae*) [113], and oat (*Avena sativa*) [114]. In addition to phenolics compounds, it was found that AA of β-glucan powder is also linked to the β-glucan monosaccharide (glucose) units participating on its molecular structure, and it may be attributed to the presence of multiple anomeric hydrogen atoms being abstracted by the free radicals [114,115]. To our best knowledge, the remaining values determined in our microencapsulated formulations with respect to bioactive compounds were not able to compare with other reports because of their absence prior to our experiments. Hence, our study can be recognized as also pioneering in this field of research.

In summary, the results of all our analyzes revealed the SD microencapsulation of β-glucan powder to be a more appropriate method as compared to the FD one which is in accordance with other studies [52,56,116,117,118,119]. In addition, Luciana Di Giorgio et al. [49] highlighted the application of this method in the processing of heat-sensitive compounds for its short drying times (from 5 to 30 s), reflecting the possibilities of a wide range of its use in the commercial industry.

## 5. Conclusions

The current study provides valuable knowledge concerning a novel approach of commercial β-glucan powder microencapsulation by FD and SD techniques using MD as a wall material. The non-encapsulated sample was compared with the microencapsulated ones in terms of the structural appearance, MC, a_w_, AA, and the content of selected biologically active substances. Our results showed that the SD microparticles of β-glucan powder exhibited a more or less spherical form of various sizes ranging from 2–3 to 20–30 µm. Their values for MC (8.90 ± 0.44%) and a_w_ (0.456 ± 0.001) were demonstrably lower as compared to the control (12.38 ± 0.39%, 0.505 ± 0.001) and FD (11.30 ± 0.28%, 0.506 ± 0.002) microencapsulated samples. Moreover, our findings revealed that the product yield of the SD sample (44.38 ± 0.55%) was significantly higher in comparison with the FD (22.97 ± 0.33%) one. The use of this drying method even led to improved contents of β-glucan (72.39 ± 0.38%), TPC (3.40 ± 0.17 mg GAE/g), and flavonoids (3.07 ± 0.29 mg QE/g). Thus, the SD microencapsulation technique seems to have a greater beneficial impact on the enhancement of the β-glucan powder yield and functional status related to bioactive compounds content which is extremely favorable for the food industry. The SD microencapsulated β-glucan powder will be included next in our bakery experiments to assess its improved quality after food processing. In such a way, its potential for practical use in the development of products with added nutritional value will be explored.

## Figures and Tables

**Figure 1 foods-11-02267-f001:**
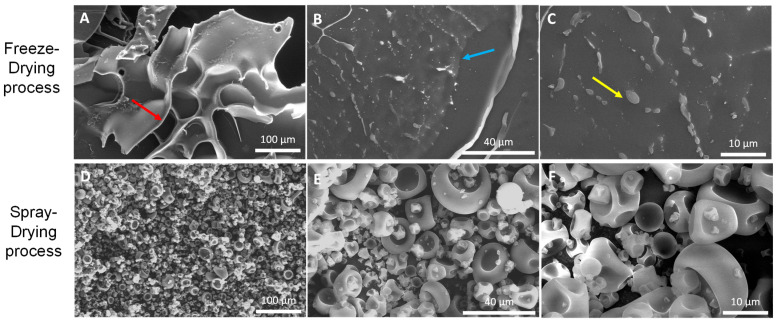
SEM images of β-glucan powder microencapsulated by: FD (**A**–**C**); and SD (**D**–**F**); methods using MD as wall material. The following magnification was applied: (**A**,**D**)—250×; (**B**,**E**)—1000×; (**C**,**F**)—2500×.

**Table 1 foods-11-02267-t001:** Determination of moisture content and water activity.

	β-Glucan Extract	Treatment 1	Treatment 2
**Moisture content (%).**	12.38 ± 0.39 ^c^	11.30 ± 0.28 ^b^	8.90 ± 0.44 ^a^
**Water activity**	0.505 ± 0.001 ^b^	0.506 ± 0.002 ^c^	0.456 ± 0.001 ^a^

Notes: Mean ± standard deviation (*n* = 3): β-glucan extract (control); Treatment 1 (freeze-drying method); Treatment 2 (spray-drying method). The lowest values for each measured parameter, as well as insignificant (*p* > 0.05) differences between the samples in the same line, are indicated by the letter “a”. Different superscript letters in the same line indicate significant differences (*p* < 0.05) between the experimental samples.

**Table 2 foods-11-02267-t002:** Determination of β-glucan content.

	β-Glucan Extract	Treatment 1	Treatment 2
**β-glucan content (%)**	58.78 ± 0.54 ^a^	70.78 ± 0.56 ^b^	72.39 ± 0.38 ^c^

Notes: Mean ± standard deviation (*n* = 3): β-glucan extract (control): Treatment 1 (freeze-drying method); Treatment 2 (spray-drying method). The lowest values for each measured parameter, as well as insignificant (*p* > 0.05) differences between the samples in the same line, are indicated by the letter “a”. Different superscript letters in the same line indicate significant differences (*p* < 0.05) between the experimental samples.

**Table 3 foods-11-02267-t003:** Determination of antioxidant activity and biological active compounds.

	β-Glucan Extract	Treatment 1	Treatment 2
**Antioxidant activity** **(%)**	14.91 ± 0.36 ^a^	14.51 ± 0.91 ^a^	14.85 ± 0.19 ^a^
**Total polyphenols content (mg GAE/g)**	2.80 ± 0.34 ^a^	3.01 ± 0.46 ^ab^	3.40 ± 0.17 ^b^
**Phenolics acids content (mg CAE/g)**	3.61 ± 0.75 ^a^	3.85 ± 0.45 ^a^	3.81 ± 0.46 ^a^
**Flavonoids content (mg QE/g)**	2.40 ± 0.35 ^a^	2.80 ± 0.36 ^ab^	3.07 ± 0.29 ^b^

Notes: Mean ± standard deviation (*n* = 3): β-glucan extract (control); Treatment 1 (freeze-drying method); Treatment 2 (spray-drying method). The lowest values for each measured parameter, as well as insignificant (*p* > 0.05) differences between the samples in the same line, are indicated by the letter “a”. Different superscript letters in the same line indicate significant differences (*p* < 0.05) between the experimental samples.

## Data Availability

Data available in a publicly accessible repository.

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
