# Peer review of "Impact of Freeze- and Spray-Drying Microencapsulation Techniques on β-Glucan Powder Biological Activity: A Comparative Study"

_foods, 2022, doi:10.3390/foods11152267_

Round 1
Reviewer 1 Report
I have reviewed the manuscript titled: Effectiveness of Freeze- and Spray-Drying Microencapsulation Techniques on β-glucan Powder Biological Activity: a Comparative Study.
This article aims to evaluate the efficiencies of freeze- and spray-drying microencapsulation ways on the content of β-glucan, total polyphenols, total flavonoids, phenolic acids, and antioxidant activity in β-glucan powder using maltodextrin as a carrier. This study found water activity and moisture content of β-glucan microencapsulated powders was shown to be low in the spray-drying powder than those of dried by freezing. The information of this work is useful and relevant and the spray-drying technique seems to be more effective in the values of the β-glucan powder stability, its contents of β-glucan, total polyphenols, and total flavonoids than the freeze-drying method. I think the manuscript is acceptable after minor revision. Although, the article is not innovative, it contains original and interesting information for food processing of bioactive compounds. Abstract is well written upon and water activity and moisture content, which can influence the shelf-life of the microencapsulated sample are mentioned and the structural differences between freeze and spray-drying microencapsulation powders are observed using scanning electron microscopy. Introduction is well addressed including intake of bioactive compounds such as β-glucan. The importance to preserve the biological activity and to keep the stability and release of the bioactive compounds were introduced. The economic advantage information of spray drying and freeze drying treatments and even disadvantage for different processing methods were mentioned.
Materials and methods were well described.
This article would be accepted if the authors revise the capital of Journal abbreviation in reference section (no. 33). Introduction, Material and methods, and Results and discussion section clearly, it will be helpful for other researchers to follow this study in the future.
I am not a native English speaker. The manuscript seems have no major mistakes are detected and the manuscript can be easily understood except the reference 33 as attached file. The results are well discussed.
I enjoyed reading this manuscript; the needs of special groups of food processing of bioactive material microencapsulation. This manuscript presents some interesting data.
Date of this review
03 July 2022 12:20

Author Response
Reviewer #1
This article aims to evaluate the efficiencies of freeze- and spray-drying microencapsulation ways on the content of β-glucan, total polyphenols, total flavonoids, phenolic acids, and antioxidant activity in β-glucan powder using maltodextrin as a carrier. This study found water activity and moisture content of β-glucan microencapsulated powders was shown to be low in the spray-drying powder than those of dried by freezing. The information of this work is useful and relevant and the spray-drying technique seems to be more effective in the values of the β-glucan powder stability, its contents of β-glucan, total polyphenols, and total flavonoids than the freeze-drying method. I think the manuscript is acceptable after minor revision. Although, the article is not innovative, it contains original and interesting information for food processing of bioactive compounds. Abstract is well written upon and water activity and moisture content, which can influence the shelf-life of the microencapsulated sample are mentioned and the structural differences between freeze and spray-drying microencapsulation powders are observed using scanning electron microscopy. Introduction is well addressed including intake of bioactive compounds such as β-glucan. The importance to preserve the biological activity and to keep the stability and release of the bioactive compounds were introduced. The economic advantage information of spray drying and freeze drying treatments and even disadvantage for different processing methods were mentioned.
Response: Thank you very much for the favorable comment.
Point 1: Materials and methods were well described.
Response: Thank you for your opinion.
Point 2: This article would be accepted if the authors revise the capital of Journal abbreviation in reference section (no. 33). Introduction, Material and methods, and Results and discussion section clearly, it will be helpful for other researchers to follow this study in the future.
Response: Revised directly in the manuscript.
Point 3: I am not a native English speaker. The manuscript seems have no major mistakes are detected and the manuscript can be easily understood except the reference 33 as attached file. The results are well discussed.
Response: Thank you for your opinion.
Point 4: I enjoyed reading this manuscript; the needs of special groups of food processing of bioactive material microencapsulation. This manuscript presents some interesting data.
Response: Thank you for your opinion.
Reviewer 2 Report
The manuscript compares two methodologies, freeze (FD) and spray drying (SD), to obtain powders rich and stable in β-glucan and other bioactive compounds. The moisture content and water activity of powders were studied. The morfology and size of microparticles were analyzed by SEM. The content of β-glucan, total polyphenols, flavonoids, and phenolic acids were determined. The antioxidant capacity was evaluated by DPPH assay.
Throughout the manuscript, the authors discuss “efficiencies” of microencapsulation, the “stability” of bioactive compounds, and the “shelf-life” of powders, but the authors have not calculated the drying yield, and the retention of bioactive compounds after microencapsulation, nor done stability studies of bioactive compounds during storage of powders. I recommend to change type of your paper from article to communication.
Please consider the following comments:
Major comments:
-English needs to be significantly improved.
-The authors used an enormous amount of published articles to write the manuscript.
TITLE
-change effectiveness, please.
ABSTRACT
-line 26: “The study compares the efficiencies of freeze- and spray-drying (FD, SD) microencapsulation methods…” The authors did not study the efficiency of microencapsulation, so use another word, please.
-line 33-35: “Values for aw and MC were shown to be more reduced (P < 0.05) in the SD microencapsulated powder as compared to that dried by freezing. The same tendency was also found for β-glucan content and biological activity…” the same tendency, therefore, β-glucan content and biological activity more reduced in SD than FD?
-line 36: use another word instead of efficient, please.
-line 39: “…powder stability…” The authors did not do storage studies.
INTRODUCTION
- What is your hypothesis? Has this been looked at before and how? What are you doing about it? Currently, you are wandering about looking to land on the aims.
-line 62-63: phenolic acids and flavonoids are families of phenolic compounds, so re-write.
-line 107-109: “To improve the biological activity of commercially obtained β-glucan powder, FD and SD microencapsulation techniques using MD as a wall material were used in the present study.” The author did not do studies to improve the biological activity of commercially β-glucan powder, the author did the study to characterize the powders obtained by FD and SD.
MATERIALS AND METHODS
-line 118: “Powder extract of β-glucan (P. ostreatus) was obtained from commercial company Dimenzia Ltd. (Kežmarok, Slovakia)”. How was produced this commercial powder (by lyophilization?)? Does it have any excipient? Which?
-line 132: “Then, active β-glucan concentrate was added in 0.5% (w/w of biopolymers)…”. 0.5% w/w of MD? Clarify what is meant by biopolymers.
-line 188-189: “ … were expressed in micrograms per g of gallic acid equivalents (GAE) dry weight (dw)” is not clear, re-write, please.
-line 188-189: “ … were expressed in micrograms per g of gallic acid equivalents (GAE) dry weight (dw)” is not clear, re-write, please.
-line 197: is not clear, re-write, please.
-line 204: is not clear, re-write, please.
-Determination of β-glucan Content: How the author do the β-glucan extraction of commercial powder and microparticles obtained by SD and FD? Please explain/clarify.
RESULTS
- Table 1, table 2, and table 3: letters must be revised since for example a for higher or lower values?
-Table 3: content in family of bioactive compounds in mg but in Materials and Methods says µg
DISCUSSION
-line 395-397: “Hence, the part of our experiment was focused on the determination of its content in all our samples in order to evaluate its retention and preservation performed by both microencapsulation techniques”. How do the authors evaluate retention and preservation?
-line 403: “This fact can indicate a better β-glucan extraction technique…” What is the β-glucan extraction technique of the authors?
-line 460: use another word instead of effective, please.
CONCLUSIONS
-line 472-473: there are no storage studies, therefore the stability in terms of MC and Aw was not demonstrably…change these lines suggesting, please.
-line 474-477: same as lines 472-473; Delete stability word, please.

Author Response
Reviewer #2
The manuscript compares two methodologies, freeze (FD) and spray drying (SD), to obtain powders rich and stable in β-glucan and other bioactive compounds. The moisture content and water activity of powders were studied. The morfology and size of microparticles were analyzed by SEM. The content of β-glucan, total polyphenols, flavonoids, and phenolic acids were determined. The antioxidant capacity was evaluated by DPPH assay.
Throughout the manuscript, the authors discuss “efficiencies” of microencapsulation, the “stability” of bioactive compounds, and the “shelf-life” of powders, but the authors have not calculated the drying yield, and the retention of bioactive compounds after microencapsulation, nor done stability studies of bioactive compounds during storage of powders. I recommend to change type of your paper from article to communication.
Response: The data obtained from product yield determination was added directly to the manuscript. Hence, we would like to keep the paper as the Research Article. The evaluation of product stability during storage will be the goal of our next study dealing with the powder addition to formulation of bakery products. In these products, their qualitative and quantitative parameters including their shelf-life will be consequently assessed.
Please consider the following comments:
Major comments:
Point 1: English needs to be significantly improved.
Response: Edited directly in the manuscript.
Point 2: The authors used an enormous amount of published articles to write the manuscript.
Response: Indeed, an enormous number of references has been used to write the paper in order to highlight the fact that the encapsulation issue has been studied to a great extent. Most importantly, the issue of beta-glucan encapsulation (as a core material) cannot be found elsewhere in the published literature. Thus, our manuscript seems to be pioneering in the collection of such data.
TITLE
Point 3: change effectiveness, please.
Response: Edited directly in the manuscript.
ABSTRACT
Point 4: -line 26: “The study compares the efficiencies of freeze- and spray-drying (FD, SD) microencapsulation methods…” The authors did not study the efficiency of microencapsulation, so use another word, please.
Response: Edited directly in the manuscript
Point 5: -line 33-35: “Values for aw and MC were shown to be more reduced (P < 0.05) in the SD microencapsulated powder as compared to that dried by freezing. The same tendency was also found for β-glucan content and biological activity…” the same tendency, therefore, β-glucan content and biological activity more reduced in SD than FD?
Response: Edited directly in the manuscript.
Point 6: -line 36: use another word instead of efficient, please.
Response: Edited directly in the manuscript.
Point 7: -line 39: “…powder stability…” The authors did not do storage studies.
Response: Edited directly in the manuscript.
INTRODUCTION
Point 8: - What is your hypothesis? Has this been looked at before and how? What are you doing about it? Currently, you are wandering about looking to land on the aims.
Response: The aim of the current study was more precisely clarified in the Introduction. In effect, taking into account the effectiveness of microencapsulation techniques in general, and many advantages and disadvantages of the individual microencapsulation methods, selection of the best one for defined experimental approaches is decisive. In our laboratories, innovative bakery products with added values are developed (e.g., Valková et al., 2020; Valková et al., 2021; Valková et al., 2022). To innovate our research activities and to enhance biological activity of incorporated plant (bioactive) materials to bakery product formulations, new approaches such as microencapsulation techniques have been implemented. At first, however, it is necessary to select the most appropriate drying method for each bioactive powder being incorporated. Consequently, developed bakery products with the microencapsulated bioactive powder will be evaluated in our laboratories from both qualitative and quantitative aspects.
Valková, V., Ďúranová, H., Štefániková, J., Miškeje, M., Tokár, M., Gabríny, L., ... & Kačániová, M. (2020). Wheat bread with grape seeds micropowder: Impact on dough rheology and bread properties. Applied Rheology, 30(1), 138-150.
Valková, V., Ďúranová, H., Miškeje, M., Ivanišová, E., Gabriny, L., & Kačániová, M. (2021). Physico-chemical, antioxidant and microbiological characteristics of bread supplemented with 1% grape seed micropowder. Journal of Food & Nutrition Research, 60(1).
Valková, V., Ďúranová, H., Havrlentová, M., Ivanišová, E., Mezey, J., Tóthová, Z., Gabríny, L., & Kačániová, M. (2022). Selected Physico-Chemical, Nutritional, Antioxidant and Sensory Properties of Wheat Bread Supplemented with Apple Pomace Powder as a By-Product from Juice Production. Plants, 11(9), 1256.
Point 9: -line 62-63: phenolic acids and flavonoids are families of phenolic compounds, so re-write.
Response: Edited directly in the manuscript.
Point 10: -line 107-109: “To improve the biological activity of commercially obtained β-glucan powder, FD and SD microencapsulation techniques using MD as a wall material were used in the present study.” The author did not do studies to improve the biological activity of commercially β-glucan powder, the author did the study to characterize the powders obtained by FD and SD.
Response: Edited directly in the manuscript.
MATERIALS AND METHODS
Point 11: -line 118: “Powder extract of β-glucan (P. ostreatus) was obtained from commercial company Dimenzia Ltd. (Kežmarok, Slovakia)”. How was produced this commercial powder (by lyophilization?)? Does it have any excipient? Which?
Response: As the β-glucan (P. ostreatus) powder is a commercial product, its production process is subject to the know-how of the manufacturer. Therefore, it is not possible to describe it in the paper. We do apologize for it. However, the extraction was achieved using extraction reagents, and the obtained extract was freely dried.
Point 12: -line 132: “Then, active β-glucan concentrate was added in 0.5% (w/w of biopolymers)…”. 0.5% w/w of MD? Clarify what is meant by biopolymers.
Response: The sentence has been modified directly in the manuscript.
Point 13: -line 188-189: “ … were expressed in micrograms per g of gallic acid equivalents (GAE) dry weight (dw)” is not clear, re-write, please.
Response: Edited directly in the manuscript.
Point 14: -line 188-189: “ … were expressed in micrograms per g of gallic acid equivalents (GAE) dry weight (dw)” is not clear, re-write, please.
Response: Edited directly in the manuscript.
Point 15: -line 197: is not clear, re-write, please.
Response: Edited directly in the manuscript.
Point 16: -line 204: is not clear, re-write, please.
Response: Edited directly in the manuscript.
Point 17: -Determination of β-glucan Content: How the author do the β-glucan extraction of commercial powder and microparticles obtained by SD and FD? Please explain/clarify.
Response: Determination of β-glucan was performed according to the recommendation of the Megazyme kit, which was used in our study. We apologize but we are not able to provide β-glucan extraction procedure because of the know-how of the manufacturer.
RESULTS
Point 18: - Table 1, table 2, and table 3: letters must be revised since for example a for higher or lower values?
Response: Directly revised in the manuscript.
Point 19: -Table 3: content in family of bioactive compounds in mg but in Materials and Methods says µg
Response: Thanks for the heads up. The unit was modified in Materials and Methods. The data are modified directly in the manuscript.
DISCUSSION
Point 20: -line 395-397: “Hence, the part of our experiment was focused on the determination of its content in all our samples in order to evaluate its retention and preservation performed by both microencapsulation techniques”. How do the authors evaluate retention and preservation?
Response: The retention (preservation) of the β-glucan content in both microencapsulated powders was assessed based on the comparison of their values for content of β-glucan with the value of the non-encapsulated (control) sample. In such a way, the impacts of both drying techniques on the parameter evaluated were investigated.
Point 21: -line 403: “This fact can indicate a better β-glucan extraction technique…” What is the β-glucan extraction technique of the authors?
Response: Again, the extraction technique of the commercial β-glucan powder is subject to the know-how of the manufacturer, therefore it is not described in detail in the manuscript. The statement “This fact can indicate a better β-glucan extraction technique” is mentioned only as a hypothesis resulting from the comparison of oud data with published studies.
Point 22: -line 460: use another word instead of effective, please.
Response: Edited directly in the manuscript.
CONCLUSIONS
Point 23: -line 472-473: there are no storage studies, therefore the stability in terms of MC and Aw was not demonstrably…change these lines suggesting, please.
Response: Edited directly in the manuscript.
Point 24: -line 474-477: same as lines 472-473; Delete stability word, please.
Response: Edited directly in the manuscript.
Reviewer 3 Report
Dear Editor and authors,
The manuscript (Effectiveness of Freeze- and Spray-Drying Microencapsulation Techniques on β-glucan Powder Biological Activity: a Comparative Study) had good idea but it needs several corrects and modifications.
1- The Abstract of manuscript needs to add some results numbers.
2- The introduction needs to add new references.
3-Several working methods lack method references which could be comfort to refer to and evaluate it. example Preparation of Biopolymer Solutions for β-glucan Concentrate Encapsulation, Freeze-drying Method, .........etc.
4-Several method is unclear and it needs a standard curve. example Phenolics Acids Content and Flavonoids Content.
5-The figure 1 and 2 needs some illustrations to show the difference between the pictures.
6-Table 1, 2 and 3, the control sample was not mentioned in the working methods chapter, while we find in the results chapter a control sample.
7- The conclusions of the manuscript content some results, please rewrite again.

Author Response
Reviewer #3
Dear Editor and authors,
The manuscript (Effectiveness of Freeze- and Spray-Drying Microencapsulation Techniques on β-glucan Powder Biological Activity: a Comparative Study) had good idea but it needs several corrects and modifications.
Response: Thank you very much for the favorable comment.
Point 1: The Abstract of manuscript needs to add some results numbers.
Response: Edited directly in the manuscript.
Point 2: The introduction needs to add new references.
Response: Thank you for your comment; however, based on the recommendation of another reviewer to reduce the number of references, we have added only one more study.
Point 3: Several working methods lack method references which could be comfort to refer to and evaluate it. example Preparation of Biopolymer Solutions for β-glucan Concentrate Encapsulation, Freeze-drying Method, .........etc.
Response: References have been added directly to the manuscript.
Point 4: Several method is unclear and it needs a standard curve. example Phenolics Acids Content and Flavonoids Content.
Response: Standard curves were added directly to the manuscript.
Point 5: The figure 1 and 2 needs some illustrations to show the difference between the pictures.
Response: Figures 1 and 2 have been modified according to the reviewer’s comment. Some redundant SEM images have been removed and colored arrows have been added to illustrate the structural features mentioned in the paper.
Point 6: Table 1, 2 and 3, the control sample was not mentioned in the working methods chapter, while we find in the results chapter a control sample.
Response: Edited directly in the manuscript.
Point 7: The conclusions of the manuscript content some results, please rewrite again.
Response: Edited directly in the manuscript.
Round 2
Reviewer 2 Report
The manuscript improved.
Please consider the following minor comments:
Lines 463-466: “Similarly, significantly lower values 463 for both physical parameters of SD samples of Eugenia dysenterica DC. fruit extract powder 464 as compared to those produced by the FD microencapsulation have also been determined 465 in the study by Daza et al. [8088].” Difficult to understand, re-write, please.
Line 473:… simultaneously considerably…: revise English
Line 488:… to evaluate its retention and preservation performed by both microencapsulation tech-…delete preservation, please.
Line 515: …is commonly widely…revise English
Author Response
Reviewer #2
The manuscript improved.
Response: Thank you for your opinion.
Please consider the following minor comments:
Point 1: Lines 463-466: “Similarly, significantly lower values 463 for both physical parameters of SD samples of Eugenia dysenterica DC. fruit extract powder 464 as compared to those produced by the FD microencapsulation have also been determined 465 in the study by Daza et al. [8088].” Difficult to understand, re-write, please.
Response: Edited directly in the manuscript.
Point 2: Line 473:… simultaneously considerably…: revise English
Response: Edited directly in the manuscript.
Point 3: Line 488:… to evaluate its retention and preservation performed by both microencapsulation tech-…delete preservation, please.
Response: Edited directly in the manuscript.
Point 4: Line 515: …is commonly widely…revise English
Response: Edited directly in the manuscript.
Reviewer 3 Report
Dear Editors,
The authors made all the necessary changes to improve the manuscript, and now I recommend it for publication in its current form.
Best
Author Response
Reviewer #3
Dear Editors,
The authors made all the necessary changes to improve the manuscript, and now I recommend it for publication in its current form.
Best
Response: Thank you for your opinion.